# Machine Learning Assisting the Prediction of Clinical Outcomes following Nucleoplasty for Lumbar Degenerative Disc Disease

**DOI:** 10.3390/diagnostics13111863

**Published:** 2023-05-26

**Authors:** Po-Fan Chiu, Robert Chen-Hao Chang, Yung-Chi Lai, Kuo-Chen Wu, Kuan-Pin Wang, You-Pen Chiu, Hui-Ru Ji, Chia-Hung Kao, Cheng-Di Chiu

**Affiliations:** 1Spine Center, China Medical University Hospital, Taichung 404327, Taiwan; d30392@mail.cmuh.org.tw (P.-F.C.); assocs000@gmail.com (Y.-P.C.); taurus820514@gmail.com (H.-R.J.); 2Department of Neurosurgery, China Medical University Hospital, Taichung 404327, Taiwan; 3Department of Electrical Engineering, National Chung Hsing University, Taichung 40227, Taiwan; chchang@nchu.edu.tw; 4Department of Nuclear Medicine and PET Center, China Medical University Hospital, Taichung 404327, Taiwan; daniellai999@hotmail.com (Y.-C.L.); d10040@mail.cmuh.org.tw (C.-H.K.); 5Center of Artificial Intelligence, China Medical University Hospital, Taichung 404327, Taiwan; robinsixrainbow@gmail.com (K.-C.W.); a35374@mail.cmuh.org.tw (K.-P.W.); 6Graduate Institute of Biomedical Electronics and Bioinformatics, National Taiwan University, Taipei 10617, Taiwan; 7Department of Computer Science and Engineering, National Chung Hsing University, Taichung 40227, Taiwan; 8School of Medicine, China Medical University, Taichung 404327, Taiwan; 9Graduate Institute of Biomedical Science, China Medical University, Taichung 404327, Taiwan; 10Department of Bioinformatics and Medical Engineering, Asia University, Taichung 41354, Taiwan; 11Graduate Institute of Medical Sciences, National Defense Medical Center, Taipei 11490, Taiwan

**Keywords:** degenerative disc disease, radiomics, machine learning, T2W image, lumbar spine, low back pain

## Abstract

Background: Lumbar degenerative disc disease (LDDD) is a leading cause of chronic lower back pain; however, a lack of clear diagnostic criteria and solid LDDD interventional therapies have made predicting the benefits of therapeutic strategies challenging. Our goal is to develop machine learning (ML)–based radiomic models based on pre-treatment imaging for predicting the outcomes of lumbar nucleoplasty (LNP), which is one of the interventional therapies for LDDD. Methods: The input data included general patient characteristics, perioperative medical and surgical details, and pre-operative magnetic resonance imaging (MRI) results from 181 LDDD patients receiving lumbar nucleoplasty. Post-treatment pain improvements were categorized as clinically significant (defined as a ≥80% decrease in the visual analog scale) or non-significant. To develop the ML models, T2-weighted MRI images were subjected to radiomic feature extraction, which was combined with physiological clinical parameters. After data processing, we developed five ML models: support vector machine, light gradient boosting machine, extreme gradient boosting, extreme gradient boosting random forest, and improved random forest. Model performance was measured by evaluating indicators, such as the confusion matrix, accuracy, sensitivity, specificity, F1 score, and area under the receiver operating characteristic curve (AUC), which were acquired using an 8:2 allocation of training to testing sequences. Results: Among the five ML models, the improved random forest algorithm had the best performance, with an accuracy of 0.76, a sensitivity of 0.69, a specificity of 0.83, an F1 score of 0.73, and an AUC of 0.77. The most influential clinical features included in the ML models were pre-operative VAS and age. In contrast, the most influential radiomic features had the correlation coefficient and gray-scale co-occurrence matrix. Conclusions: We developed an ML-based model for predicting pain improvement after LNP for patients with LDDD. We hope this tool will provide both doctors and patients with better information for therapeutic planning and decision-making.

## 1. Introduction

Lumbar degenerative disc disease (LDDD) is a leading cause of lower back pain [1,2] and often serves as the first indicator of lumbar spinal degeneration, which will progress to lumbar disc herniation, spurs formation, spinal stenosis, and finally degenerative spondylolisthesis [3,4]. The ideal goals of LDDD treatment are both relieving symptoms and postponing the degenerative process [5]. To achieve these purposes, current therapeutic approaches to LDDD often involve a stepwise manner, which starts from conservative treatment (medication and rehabilitation); interventional intradiscal therapies (IDTs) such as lumbar nucleoplasty (LNP) and cooled radiofrequency ablation; discectomy; and ends with the last solution of fusion fixation surgery [6,7]. IDTs represent a critically important shift from conservative approaches to open surgery, and the efficacy of IDTs is the determinantal factor of whether to adopt the next step of LDDD management [8]. Thus, a modality that is able to predict the prognosis of IDTs is likely to be beneficial both in helping patients to decide when/whether to receive IDTs and in helping doctors to plan therapeutic strategies [8]. In response to the increasing clinical need to accurately predict pain and functional outcomes following lumbar surgery, several authors have attempted to develop models to approach these requirements [9,10,11,12,13,14,15,16,17]. For example, several previous studies have discussed how machine learning can aid in the detection and classification of LDDD using magnetic resonance imaging (MRI) [14,15]. Oktay et al. even claimed that with the use of machine learning in MRI, the diagnosis of LDDD can achieve an accuracy rate of 92.8% [13]. However, these studies only focused on diagnosis and grading without addressing clinical solutions or prognosis. A well-known report proposed a segmentation network for diagnosing LDDD from T2-weighted images (T2WIs) and established an automated quantitation system for LDDD evaluation to improve diagnostic accuracy and efficiency [16]. Nevertheless, this model also did not provide suitable suggestions for further treatment or outcome prediction. D’Antoni et al. discussed the current state of artificial intelligence (AI) in diagnosing chronic low back pain but did not focus on the decisive roles of intervertebral discs in lower back pain [17]. In addition, Wirries et al. explored how AI can facilitate decision-making for the treatment of lumbar disc herniation [12]. Staartjes et al. first investigated the pre-operative prediction of prognosis after lumbar discectomy [18]. However, both studies only discussed the choices between and results of conservative treatment and discectomy but omitted the crucial role of intervertebral discs.

MRI is a non-invasive method and represents the gold standard for detecting and diagnosing LDDD. Axial T2WI during lumbar MRI (L-MRI) improves tissue contrast and provides higher sensitivity than traditional computerized tomography (CT) imaging. Axial T2WI is used as an image target to provide richer information on the status of the nucleus pulposus of the intervertebral disc; the nucleus pulposus is mainly composed of 66% to 86% water, with the remainder consisting primarily of type II collagen and proteoglycans. The matrix of cartilage (soft bone) is made up of glycosaminoglycans, proteoglycans, collagen fibers, and sometimes elastin. The similarities in materials between the intervertebral disc and soft bone construct the intervertebral fibrocartilaginous joint. To better interpret the differences between them and surrounding tissues in various characteristics, such as collagen content, texture characteristics, annular tearing, degree of degeneration, flavum ligament status, nerve impingement, canal stenosis, soft bone, and porosity tissue, T2WI axial MRI is most helpful. The differentiation of these tissues in image presentation is relevant to the improvement of patients’ surgical pain [19,20,21]. Therefore, L-MRI is a routine imaging examination performed prior to lumbar spinal surgery.

Radiomic feature analysis refers to a quantitative approach for converting images into parameters that can be used in advanced mathematical analyses, and it has been adopted for a wide range of medical applications, including the evaluation and interpretation of radiologic and pathologic images [22]. To analyze and quantify image textures, the AI-assisted mathematical extraction of spatial distributions for signal intensities and inter-pixel relationships has been effectively applied to the interpretation of medical images [23]. The term “texture” originates from the surface characterization of textiles and can be used to describe the arrangement of any material composition visible in medical imaging. For example, the lung or blood vessel texture is the most important property and concept bridge to personalized medicine [24,25]. However, no unified or perfect mathematical model currently exists that can be applied to all medical needs.

The technological analysis of medical imaging has rapidly advanced over recent years [26], accompanied by an increase in the scale of available data, resulting in an increased interest in the field of medical radiomics [27]. High-throughput calculations are used to extract quantitative features from standard medical images, such as tomographic images (CT, MRI, positron emission tomography) [28]. Radiomics is applied in many fields through the numerical quantitative analysis of image intensity, texture, or shape, minimizing the subjectivity of image reading and reducing inaccurate interpretations and misdiagnoses [23,25,29].

The present study represents the first attempt at developing a predictive model using AI and radiomic feature analysis for LNP prognosis in patients with LDDD. The model provides predictions for therapeutic prognosis to support clinicians’ decision-making about treatment strategies for LDDD. In addition, the model also has the potential to reduce patients’ unnecessary suffering resulting from treatment delays. It could also lead to a more personalized approach to patient care, optimized treatment plans, and ultimately improved patient outcomes.

## 2. Materials and Methods

### 2.1. Population

This retrospective study enrolled 181 patients diagnosed with LDDD who received LNP. Information was obtained from the China Medical University Hospital Information System, including pre-operative L-MRI results, sex, age, height, weight, body mass index (BMI), and medical history data, such as perioperative medical and surgical details and clinical outcomes.

The exclusion criteria included patients younger than 20 years and patients lacking L-MRI data within 180 days before the operation. A total of 181 de-identified patients (associated with 314 axial T2WIs) were enrolled in this study (Figure 1). The data used in this project were approved by the Institutional Review Board of China Medical University Hospital under certificate number CMUH109-REC3-033. All patients had signed an informed consent form with a clear comprehension of the study details.

### 2.2. Primary Outcomes

The outcome of interest was the visual analog scale (VAS) value, including the values recorded within 1 month before surgery (pre-VAS) and 1 month after surgery (post-VAS). In addition, sex, age, height, weight, BMI, and disc level were recorded (Table 1). All axial T2WIs from L-MRI were collected from the picture archiving and communication system (PACS). This study compared the differences between VAS values before and after surgery (dVAS), calculated as dVAS = post-VAS − pre-VAS. An improvement rate (improvement rate (IR) = −dVAS/pre-VAS) greater than or equal to 80% (IR ≥ 80%) was defined as a significant improvement; all other values were defined as a non-significant improvement. We divided all subjects into two groups based on the IR: group A contained patients who showed significant improvement, and group B contained patients who showed non-significant improvement. In the final analysis, group A included 84 patients, and group B included 97 patients (Table 2).

### 2.3. MRI Parameters

All L-MRI data were performed within 180 days before operation. These images were acquired by two 1.5T MR scanners (Signa Excite 1.5T MR system, 71 persons; Toshiba MRT 200 MR system, 150 persons) in China Medical University Hospital. The parameters loaded from these two MRIs were the following: (1) Signa Excite 1.5T (GE Healthcare, IL, USA) with the following parameters: axial T2-weighted frFSE (TR range/TE range, 4800/114.66; echo train length, 15; bandwidth, 81 kHz; matrix, 320 × 224; slice thickness, 4 mm; and Percent Phase FOV, 100. (2) Toshiba MRT 200 1.5T (Canon Medical Systems, CA, USA) with the following parameters: axial T2-weighted (TR range/TE range, 5600/120; echo train length, 23; matrix, 320 × 192; slice thickness, 4 mm; and Percent Phase FOV, 100.

### 2.4. Data Preprocessing

In addition to image information, physiological patient data were used to develop the machine learning (ML) models, including sex, age, operative level, pre-VAS, and post-VAS (Figure 2). Prior to generating the ML models, L-MRI axial-T2WI and clinical physiological parameters underwent data preprocessing, which involved the following five steps (Figure 3).

#### 2.4.1. Rescale

Image size varied as images were acquired using multiple MRI scanners. The original DICOM image pixel sizes were 320 × 320, 320 × 356, 512 × 512, and 512 × 568. We collected the pixel spacing value in every DICOM file and counted the plural of these figures. We rescaled all images to display at the same scale by adjusting the images based on the pixel spacing value (the plural was approximately 0.35). After rescaling, the image size changed to 511 × 511, 512 × 512, 540 × 540, 568 × 511, 568 × 512, 568 × 568, 625 × 625, and 654 × 654. These images were obtained from multiple MRI scanners, and differences in image scaling can affect feature calculations. We then extended outwards from the critical hot spots of disc degeneration and cropped the images to a size of 224 × 224 to facilitate focused calculation of critical features in subsequent analyses. We used the radiomic toolkit to extract features from the images, including descriptors of the intensity histogram, size- and shape-based features, voxel relationship descriptors, e.g., size zone matrix (SZM), gray-level co-occurrence matrix (GLCM), etc.; there were 91 features. Finally, we standardized the clinical and radiomic feature data by transforming them from their natural range (e.g., age range from 1 to 100) into a standardized range of 0 to 1.

#### 2.4.2. Crop

In order to focus on disc degeneration of the lumbar spine. Our study cropped the MRI image size to 224 × 224 by extension from the image center, which contained the critical hot spots of disc degeneration.

#### 2.4.3. Radiomic Features Extraction

Radiomic features can be classified into five categories: descriptors of the intensity histogram, size, and shape-based features, voxel relationship descriptors [e.g., SZM, GLCM, run length matrix (RLM), neighborhood gray-tone difference matrix (NGTDM)-derived textures, and filtered image], and fractal-based textures. In this study, 91 radiomic features were extracted from these five categories—please see (Table 3).

#### 2.4.4. Normalization

Normalization is widely known to improve the performance and training stability of the model. Our study converted clinical data and radiomic feature data from their natural range (for example, age range from 1 to 100) into a standard range of 0 and 1.

#### 2.4.5. K-Fold (Cross-Validation)

To avoid model training bias, we take the K-fold cross-validation widely used to abstain the specific data separation in the training and testing dataset. First, split the training dataset into K pieces, taking one piece as the validation set each time, and the remaining data as the training set, so we can obtain the results of K validation sets. The results obtained by averaging multiple test sets are often more representative. The advantage of this step is that when there is not much training data, the model’s generalization ability is tested multiple times in the limited training set, and the results obtained by averaging over multiple validation sets are usually more representative. When separating the data, we avoided repeated patient data from being mixed in the training and test sets.

### 2.5. Model Development and Validation

#### 2.5.1. Support Vector Machine (SVM)

SVM maps the training data to the hyperplane to maximize the margin of the gap between the two categories. The farther the classification boundary is from the nearest training data point, the better because this can reduce the generalization error of the classifier. In cases where the data is not linearly separable, we need to transform the data to a higher dimensional space to accommodate the support vector classifier.

#### 2.5.2. Light Gradient Boosting Machine (LGBM)

LGBM uses a histogram algorithm to segment continuous features to speed up the training process and reduce memory load. In addition, LBGM takes advantage of the leaf-wise best-first method to determine DT growth, which reduces losses and avoids overfitting.

#### 2.5.3. Extreme Gradient Boosting (XGB)

XGB is a DT-based boosting algorithm that utilizes a new generalized gradient boosting DT algorithm to speed up model construction.

#### 2.5.4. Extreme Gradient Boosting Random Forest (XGBRF)

XGBRF is an ensemble learning method for classification, regression, and other tasks, which operates by constructing a multitude of decision trees (DTs) during training. The advantage of this algorithm is the ability to accommodate a high noise margin and high-dimensional feature data.

#### 2.5.5. Categorical Boosting (CatBoost)

CatBoost is a gradient boosting library based on binary decision trees. Target leakage and prediction shifts are avoided by grouping categories with target statistics (TS). The log loss and zero-one loss were better than the traditional greedy algorithm.

#### 2.5.6. Improved Random Forest (iRF)

The random forest used in this study underwent some optimizations. Random forest is an ensemble machine learning algorithm that uses multiple decision trees to make predictions. Each decision tree in the random forest is trained on a random subset of the available data and features. The final prediction is made by aggregating the predictions of all the individual trees. The essential advantage of the random forest algorithm is that it reduces the risk of overfitting by introducing randomness to the training process.

In this study, we used optimization parameters to control the leaves and depth of the decision trees. They experimented with different values of the maximum depth of the trees, and a minimum number of samples were required to split an internal node. These parameters have a significant impact on the performance of the random forest model as they control the complexity of the decision trees and the degree of overfitting.

We used a cross-validation approach to select the optimal values of these parameters. We split the available data into training and validation sets. We then trained multiple random forest models on different subsets of the training data set with different combinations of the optimization parameters. The models were then evaluated using the validation data set, and the combination of parameters that performed the best was selected as the optimal configuration.

In addition to the optimization parameters, we used a random seeding mechanism to choose the best model after a certain number of random seeds. Random seed ensures that the random forest model is trained on a different subset of the data and features each time it is run, thereby introducing additional randomness to the training process. By running the model multiple times with different random seeds and selecting the best-performing model, we ensured that the final model was not overly sensitive to the initial random seed and was robust to variations in the training data. Overall, the combination of optimization parameters and the random seeding mechanism used in the study allowed us to build a high-performing random forest model robust to overfitting and variations in the training data (Figure 4).

### 2.6. Model Training Parameters

Most of the models were generated using default parameters in Scikit-learn. The SVM algorithm used radial based function kernel, the complexity of model was 100, and the gamma was 0.6. For the LGBM algorithm, the number of DTs was 202, the maximum tree depth was 40, learning rate was 0.05, the number of leaves was 25, and the proportion of features used when built per tree was 0.53. For the XGB algorithm, 305 DTs were used, the depth was 29, learning rate was 0.6, the number of leaves was 46, and the proportion of features used when built per tree was 0.52. For the XGBRF algorithm, the number of DTs was 451, the maximum tree depth was 29, learning rate was 1, the number of leaves was 46, and the proportion of features used when built per tree was 0.52. The iRF algorithm was implemented using MATLAB; the maximum tree number was 20, the number of leaves was 20, and the learning rate was 1.

### 2.7. Model Training Equipment

All models were trained on the NVidia DGX-2 system (NVIDIA, CA, USA): central processing unit, Dual Intel Xeon Platinum 8168 (Intel, CA, USA), 2.7 GHz, 24-cores × 2; system memory, 1.5 TB RAM; storage, 960 GB solid-state drive (SSD) × 2, 30 TB internal SSD; graphics processing unit (GPU), 16-slice NVIDIA Tesla V100 (NVIDIA, CA, USA); GPU memory, 512 GB; software: Ubuntu Linux operating system (OS, version 18.04.3), Red Hat Enterprise Linux OS (version 9.2); the total training time was about 12 h. The software versions and environment packages are detailed as follows: python 3.10.9, pandas 1.5.3, tensorflow 2.12.0, matplotlib 3.7.0, scikit-learn 1.2.1.

### 2.8. Evaluation Index

To ensure the fairness of the model, the performance of each model was evaluated using the same testing set. We split the modeling dataset into training and testing sets using the K-fold cross-validation method, which assigns 80% of data points to the training set and the remaining 20% of data points to the testing set. As mentioned above, we avoided repeated patient data from being mixed in the training and test sets. Evaluation indicators, including the confusion matrix, accuracy, sensitivity, specificity, F1 score, and area under the receiver operating characteristic (ROC) curve (AUC), were used to measure model performance. The Scikit-learn suite was used to establish the derivation among evaluation parameters and verify each model.
(1)$$Accuracy=\frac{TP+TN}{TP+FP+TN+FN}$$
(2)$$Sensitivity=\frac{TP}{TP+FN}$$
(3)$$F1\;score=2*\frac{Recall*Precision}{Recall+Precision}$$
(4)$$Specificity=\frac{TN}{FP+TN}$$
(5)$$Precision=\frac{TP}{TP+FP}$$
(6)$$Recall=\frac{TP}{TP+FN}$$

*TP*, True Positive; *TN*, True Negative; *FP*, False Positive; *FN*, False Negative.

## 3. Results

### 3.1. Models

This study developed various ML models for predicting pain relief following LNP based on the pre-operative L-MRI T2WI and physiological parameters of patients with LDDD and verified the predictive efficacy of each developed model. Physiological parameters included sex, age, disc level, and VAS. Both L-MRI T2WI and physiological data were preprocessed before model training. Six types of ML models were trained and compared, including SVM, LGBM, XGB, and XGB, based on RF and iRF, and the results are shown in (Table 4). The best-performing model was the iRF model, which had an accuracy of 0.76 for outcome prediction. CatBoost was the next-best-performing model, with an accuracy of 0.63. It was the best-performing model for sensitivity (0.70). iRF’s sensitivity was close to 0.69, followed by XGB based on RF, LGBM, XGB, and SVM, which all had sensitivity values less than 0.6. The iRF model had the best performance for specificity (0.83), followed by XGB based on RF (0.62). The F1 scores of the iRF and CatBoost models were 0.73 and 0.63, respectively. We next evaluated the AUC of each model. iRF achieved an AUC of 0.77, and XGB based on RF, CatBoost, and LGBM achieved an AUC of about 0.6. The ROC curve for one data re-sampling run is shown in Figure 5. The AUCs of most models were close to 0.6, except for SVM, which had an AUC of 0.53. The AUC of iRF was 0.77, which was the closest to 1 among the AUCs of all the models tested.

### 3.2. Radiomic Features

The iRF model was the best performing in comparison with four other machine learning models. The iRF model was trained using a combination of L-MRI T2WI and physiological parameters to develop a predictive model of postoperative pain improvement.

To obtain texture information from T2WI and volumes of interest in L-MRI, the Pyradiomics suite was used to extract five sets of radiomic features, resulting in 91 radiomic features used for training the ML models.

We ranked the top five influential factors after analyzing the influences of the various features used to train the ML models (Figure 6). The pre-VAS parameter was found to have the highest weight ratio for patients with significant postoperative pain improvement (Group A) and those with non-significant improvement (Group B). The second highest weight ratio was assigned to the diagnostics image original minimum, which showed a marked disparity from the first coefficient of the pre-VAS parameter. The third most influential parameter was the maximum probability of GLCM, the fourth was the patient’s sex ranking, and the fifth was the zone variance of GLSZM.

Based on these findings, we concluded that the patient’s physiological parameters had a more significant impact on the accuracy of the ML models than radiomic features. This highlights the importance of incorporating physiological parameters into predictive models for postoperative pain improvement. Overall, the iRF model trained using the combination of L-MRI T2WI and physiological parameters proved to be a highly accurate and effective predictive model of postoperative pain improvement.

The optimization strategy used in this study was control of the depth of the tree and the number of leaves in the iRF model. The depth of the tree refers to the number of feature segmentation points used in the decision tree, while the number of leaves determines the granularity of the tree structure. We aimed to achieve an optimal balance between model complexity and accuracy by adjusting these parameters.

Once the iRF model was trained, we used it to analyze the feature importance of the data set. We did this by computing branch prediction performance from the last leaf of each decision tree in the forest. By examining the feature segmentation points associated with the best-performing leaves, we determined the influence of each feature on overall model performance.

The iRF algorithm was particularly useful because it allowed us to identify the data set’s most essential features while accounting for their interactions with other features. We obtained a more accurate and interpretable model that provided valuable insights into the underlying data. Overall, the iRF model proved to be a robust tool for feature selection and analysis in this study.

## 4. Discussion

Our iRF model had the best performance among all evaluation indicators, with accuracy, F1 score, sensitivity, and specificity values of 0.76, 0.73, 0.69, and 0.83, respectively. Table 5 shows the confusion matrix for test sets in the iRF algorithm and the other five algorithms. iRF outperformed the other five models in the most evaluated aspects. Aging is a primary contributor to intervertebral disc degeneration, which cannot be prevented [4]. However, we can attempt to postpone the final sequela of degenerative spinal instability or reduce the symptoms and signs accompanying the natural course of degeneration to achieve a longer and better quality of life [12]. Fusion surgery is likely the final therapeutic choice when degenerative spinal instability occurs [6]. When conservative treatments fail, patients are likely to seek more advanced therapies for their back pain. Therefore, the decision and timing of IDTs play a critical role in developing stepwise therapeutic strategies [8]. The decision to pursue LNP can shorten the unnecessary painful duration and help timely intervention for preventing the deterioration of the clinical course. Thus, the importance of developing a reliable predictor of prognosis based on LNP outcomes using clinical or imaging data cannot be over-emphasized.

Traditionally, the decision of who receives or when to receive LNP relies solely on the experience and suggestions of physicians. Most prognostic and outcome assessments are based on three factors: (1) general patient data such as age, sex, and body mass index; (2) physicians’ subjective reading of images, e.g., of the positive hyperintensity zone, black disc, and Modic changes of the end plates; and (3) invasive challenge discography results [30]. Furthermore, most previous studies have only demonstrated the efficacy of LNP, but few have discussed the outcome prediction or prognostic factors [31,32,33]. With AI-assisted assessments based on patients’ general data and objective analysis of image radiomic features, we can not only prevent human errors but also reduce the invasiveness of diagnostic tools such as discography.

Klessinger et al. reported a short-term repeat surgery rate as high as 18.7% for LNP, making pre-operative assessment crucial for preventing poor outcomes [34]. However, the meta-analysis by Eichen et al. showed that LNP reduces pain in the long term and improves patients’ functional mobility [32]. Cincu et al.’s report, based on 10 year follow-up data, concluded that LNP might provide intermittent pain relief in contained disc herniation without significant complications [35]. These differences between studies may be due to variability in patient selection and decision-making [36]. Based on the therapeutic methods of LDDD, Huo et al. compared the efficacy between LNP and endoscopic discectomy and showed that LNP is more appropriate for young and middle-aged patients, a finding consistent with our results [37]. Although there are several IDTs in addition to open discectomy, their outcomes vary and may depend on the interventionalist’s expertise and careful patient selection [38]. In addition, as there are many uncertainties regarding the therapeutic outcomes of LNP, AI-assisted prediction models may provide objective references for decision-making in the medical needs assessment.

Six ML algorithms were used to develop the models, including SVM, LGBM, XGB, XGBRF, CatBoost, and iRF, and their performance was compared. As a result of comparing overall performance, including accuracy, sensitivity, specificity, F1 score, and AUC (Figure 5 and Table 4), the iRF model proved to be the most successful. In this study, SVM is an overly simplistic approach. Large variance and complex cases make LGBM and XGB less effective [39,40]. Insufficient training data may degrade the performance of XGBRF when faced with high-dimensional feature data [41]. The iRF optimizes the number of trees and leaf branches to prevent overfitting, as well as the tree number, leaf depth, and seed. This study shows that iRF has the best performance for classification problems involving a large number of features.

ML algorithms analyze data automatically and generate rules for predicting unknown data [42]. Deep learning (DL) is a branch of ML in which algorithms learn data features by using artificial neural network architecture [43]. Deep learning is capable of processing more complex data, and the performance usually increases as more data is available, allowing the network to recognize more features. As a result of the small amount of data collected in this study, DL algorithms were not able to learn effectively, which led to poor performance. In order to improve the results of deep learning algorithms, additional data or data augmentation techniques are likely to be used [44]. DL models, such as convolutional neural networks, can be used to learn the graphs for further classification after converting features into histograms or graphs [45].

According to the results of feature importance based on these influential factors (Figure 6), patients’ pre-operative VAS scores ranked first. This could indicate that patients who experience more pain before LNP have more room for subjective improvement. Furthermore, the patient’s age is also a critical pathophysiological influencing factor. This is consistent with the mechanism of LNP, in which the younger and collage-richer the pulposus nucleus contained in the intervertebral disc, the more effective the disc coblation will be [36,46,47]. Compared to the radiomic features, the overall physiological parameters had a greater impact on building the ML models, which suggests that the AI-assisted image reading played the role of an assistant rather than a replacement. On the other hand, three of the top five most significant features were directly related to L-MRI axial-T2WI image radiomic features. These outcomes indicate that the iRF ML model in our study possesses high clinical value beyond image analysis. It may serve as a reliable predictor, as outcomes were highly correlated with LNP prognosis.

## 5. Conclusions

In this study, the iRF ML model achieved the highest accuracy for predicting pain improvements in patients with LDDD following LNP. It can contribute to LDDD prognostic prediction and provide a reference for LNP decision-making. However, this retrospective study has several limitations. (i) The case number was small, resulting in the unsuccessful fitting of DL models. (ii) Uniformity of images was difficult to achieve due to differences in the MRI machines used to obtain images or in the image format. (iii) No indicators of LDDD severity were available beyond the VAS score. (iv) Individual differences in IDTs were not examined in this study. (v) Missing data might also have affected the study outcomes. It is essential to compare prospective differences between iRF ML model predictions and real-world results. Therefore, further work on model validation based on a comparison of results is required.

## Figures and Tables

**Figure 1 diagnostics-13-01863-f001:**
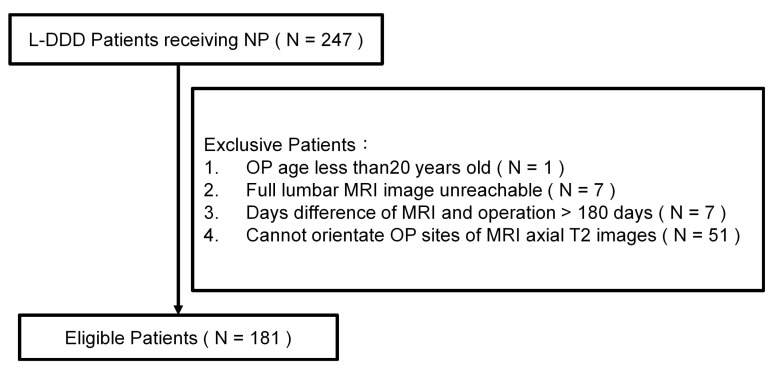
A flow chart showing the patient selection process for the present study (part I).

**Figure 2 diagnostics-13-01863-f002:**
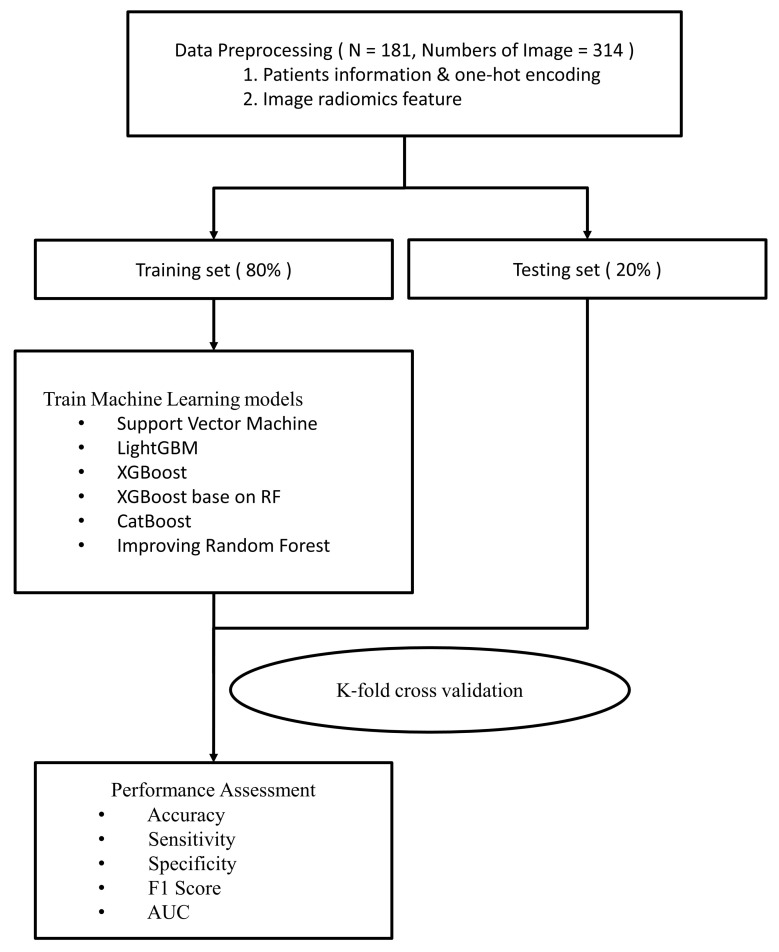
Flow chart showing the study procedures (part II).

**Figure 3 diagnostics-13-01863-f003:**
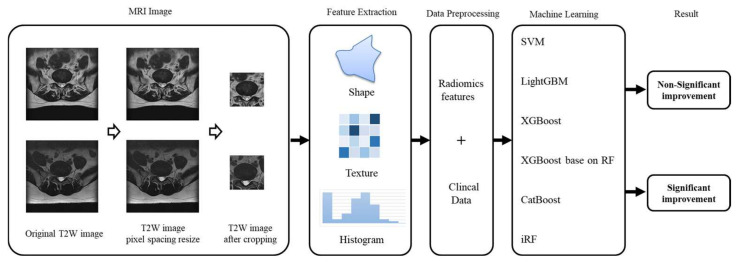
Methodology pipeline.

**Figure 4 diagnostics-13-01863-f004:**
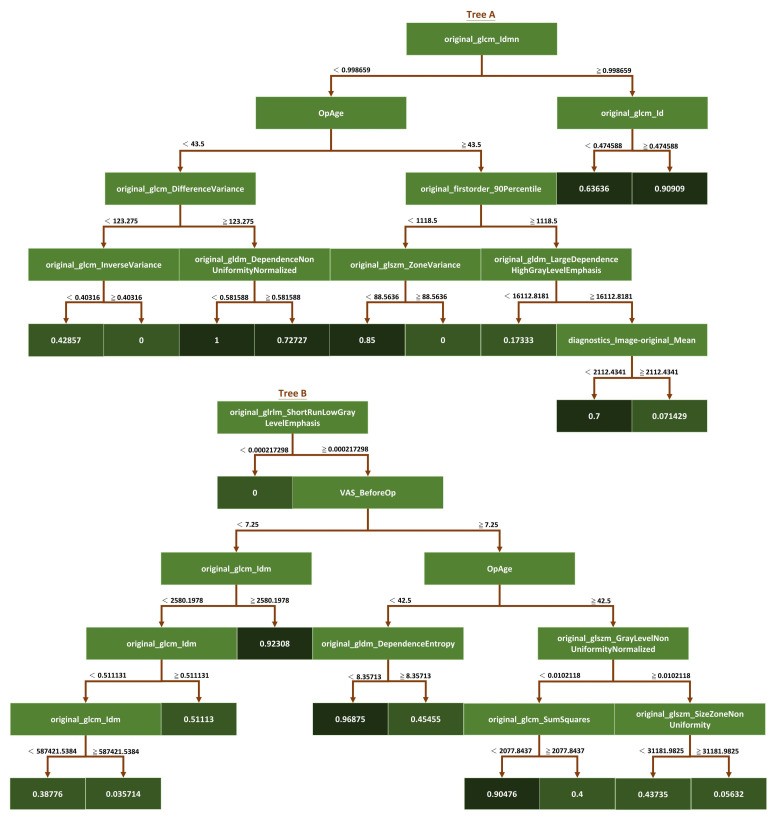
Schematic diagram of the iRF model. This study used the iRF algorithm to analyze the feature importance of a data set. The iRF model is based on the random forest algorithm, but it includes additional steps to eliminate less essential features and iteratively retrain the model. The dark green squares represent the classification results at the endpoints of each tree branch.

**Figure 5 diagnostics-13-01863-f005:**
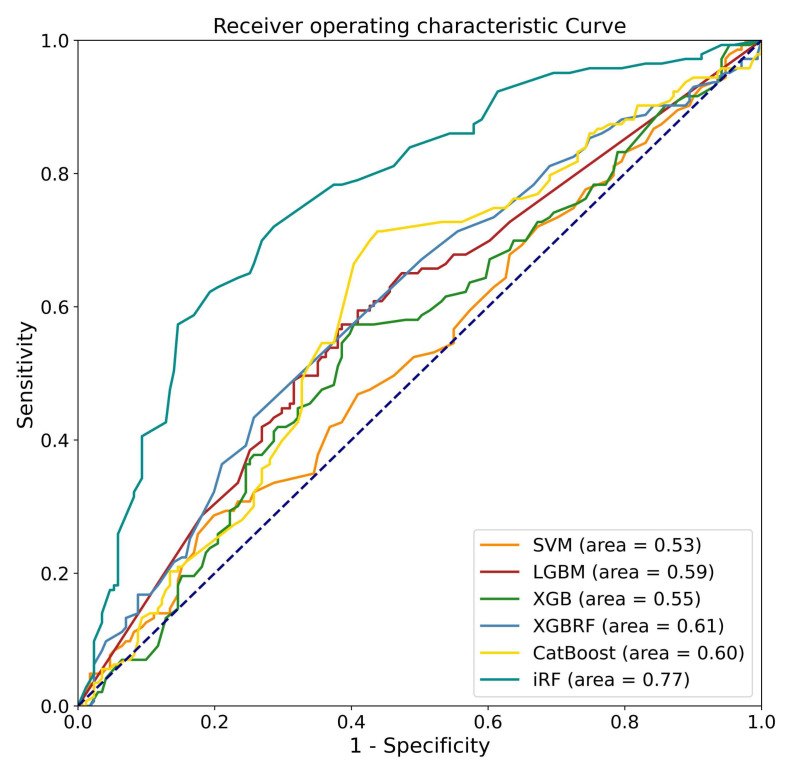
Receiver operating characteristic curves for the performance of the machine learning algorithms when analyzing the testing sets.

**Figure 6 diagnostics-13-01863-f006:**
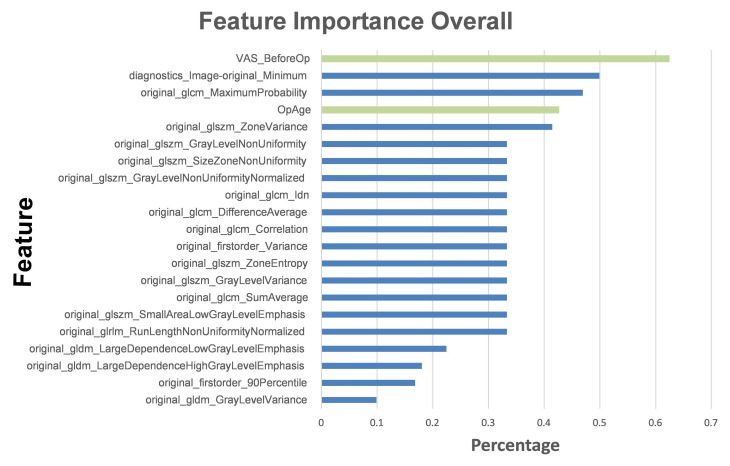
Top features used to train the iRF model. The histogram describes the proportions of feature importance for the iRF model. The green columns represent features from basic clinical characteristics, while the blue columns represent radiomic features.

**Table 1 diagnostics-13-01863-t001:** Patient’s characteristics.

Clinical Characteristics (*n* = 181)	
Male	98 (54.1%)
Age (years) *	45.4 ± 10.4
Height (cm) *^,a^	165.6 ± 8.4
Weight (kg) *^,b^	69.0 ± 12.7
BMI *^,c^	25.3 ± 3.7

BMI, body mass index. * Values are expressed as the mean ± SD. Percentage of data deficient: ^a^ Height = 17.1%, ^b^ Weight = 30.4%, ^c^ BMI = 34.3%.

**Table 2 diagnostics-13-01863-t002:** Operation characteristics.

Operation Items	Non-Significant Improvement (IR < 80%)	Significant Improvement (IR ≥ 80%)
Number of patients	97	84
Date Interval Between Image and Op (days) *	17.5 ± 23.5	17.5 ± 27.6
Op Segment Num *	1.8 ± 0.7	1.7 ± 0.7
Op Segment/MRI Axial T2 Total Num	171	143
L2/L3	12 (7.0%)	2 (1.4%)
L3/L4	24 (14.0%)	25 (17.5%)
L4/L5	76 (44.4%)	68 (47.6%)
L5/S1	59 (34.5%)	48 (33.6%)
VAS		
Before Op *	7.0 ± 1.8	7.7 ± 1.8
After Op *	3.6 ± 1.3	0.7 ± 0.5

Op, Operation; VAS, Visual Analog Scale; L, Lumbar spinal; S, Sacrum. * Values are expressed as the mean ± SD.

**Table 3 diagnostics-13-01863-t003:** Radiomic features.

Radiomic Feature Items	Non-Significant Improvement (dVAS < 80%)	Significant Improvement (dVAS ≥ 80%)	*p* Value
diagnostics_Image-original_Mean	1368.18 ± 743.97	1504.12 ± 732.85	0.406
diagnostics_Image-original_Minimum	−26.80 ± 153.84	−23.14 ± 168.39	0.310
diagnostics_Image-original_Maximum	5235.06 ± 2482.10	5612.69 ± 2411.79	0.481
original_firstorder_10Percentile	608.20 ± 371.87	677.73 ± 373.53	0.320
original_firstorder_90Percentile	2600.52 ± 1342.56	2830.89 ± 1314.54	0.333
original_firstorder_Energy	163,674,023,588.18 ± 132,128,707,225.62	188,331,818,772.32 ± 132,077,417,316.23	0.800
original_firstorder_Entropy	6.30 ± 1.04	6.47 ± 0.99	0.223
original_firstorder_InterquartileRange	1033.82 ± 606.15	1114.88 ± 629.7	0.780
original_firstorder_Kurtosis	4.57 ± 1.81	4.7 ± 1.76	0.985
original_firstorder_Maximum	5235.06 ± 2482.1	5612.69 ± 2411.79	0.481
original_firstorder_MeanAbsoluteDeviation	657.37 ± 331.50	710.06 ± 333.02	0.551
original_firstorder_Mean	1368.17 ± 743.97	1504.11 ± 732.84	0.406
original_firstorder_Median	1066.95 ± 619.43	1187.38 ± 617.89	0.863
original_firstorder_Minimum	−26.80 ± 153.84	−23.14 ± 168.39	0.310
original_firstorder_Range	5261.86 ± 2543.46	5635.84 ± 2446.62	0.436
original_firstorder_RobustMeanAbsoluteDeviation	450.48 ± 249.17	484.75 ± 256.48	0.931
original_firstorder_RootMeanSquared	1601.56 ± 837.35	1754.75 ± 824.08	0.427
original_firstorder_Skewness	1.34 ± 0.45	1.36 ± 0.46	0.788
original_firstorder_TotalEnergy	163,674,023,588.18 ± 132,128,707,225.62	188,331,818,772.32 ± 132,077,417,316.23	0.800
original_firstorder_Uniformity	0.02 ± 0.02	0.02 ± 0.02	0.248
original_firstorder_Variance	839,905.29 ± 626,984.51	957,832.26 ± 655,527.73	0.633
original_glcm_Autocorrelation	5530.37 ± 4687.35	6248.81 ± 4372.8	0.996
original_glcm_ClusterProminence	163,352,102.96 ± 173,653,483.58	205,207,752.11 ± 205,943,224.75	0.047
original_glcm_ClusterShade	543,555.56 ± 482,508.89	655,848.32 ± 571,342.03	0.092
original_glcm_ClusterTendency	5231.25 ± 3910.18	5971.74 ± 4095.13	0.626
original_glcm_Contrast	145.71 ± 110.52	161.72 ± 116.18	0.323
original_glcm_Correlation	0.95 ± 0.01	0.95 ± 0.02	0.207
original_glcm_DifferenceAverage	6.98 ± 3.76	7.43 ± 3.78	0.832
original_glcm_DifferenceEntropy	3.99 ± 1.01	4.10 ± 0.97	0.233
original_glcm_DifferenceVariance	81.62 ± 60.34	90.76 ± 63.17	0.294
original_glcm_Id	0.32 ± 0.15	0.30 ± 0.14	0.101
original_glcm_Idm	0.25 ± 0.17	0.23 ± 0.16	0.077
original_glcm_Idmn	1.00 ± 0.00	1 ± 0.00	0.320
original_glcm_Idn	0.97 ± 0.01	0.97 ± 0.01	0.356
original_glcm_Imc1	−0.28 ± 0.07	−0.28 ± 0.06	0.336
original_glcm_Imc2	0.98 ± 0.01	0.98 ± 0.01	0.354
original_glcm_InverseVariance	0.23 ± 0.13	0.21 ± 0.12	0.089
original_glcm_JointAverage	56.78 ± 32.27	62.09 ± 30.8	0.360
original_glcm_JointEnergy	0.00 ± 0.01	0.00 ± 0.01	0.569
original_glcm_JointEntropy	10.86 ± 2.08	11.14 ± 1.98	0.183
original_glcm_MCC	0.95 ± 0.01	0.95 ± 0.01	0.260
original_glcm_MaximumProbability	0.01 ± 0.02	0.01 ± 0.03	0.255
original_glcm_SumAverage	113.56 ± 64.53	124.17 ± 61.59	0.360
original_glcm_SumEntropy	7.26 ± 1.04	7.42 ± 0.99	0.235
original_glcm_SumSquares	1344.24 ± 1003.55	1533.37 ± 1049.85	0.627
original_glrlm_GrayLevelNonUniformity	882.08 ± 595.29	793.96 ± 546.74	0.174
original_glrlm_GrayLevelNonUniformityNormalized	0.02 ± 0.02	0.02 ± 0.02	0.170
original_glrlm_GrayLevelVariance	1356.41 ± 1003.47	1544.7 ± 1048.67	0.648
original_glrlm_HighGrayLevelRunEmphasis	5689.55 ± 4760.94	6421.58 ± 4446.22	0.976
original_glrlm_LongRunEmphasis	1.64 ± 0.86	1.58 ± 1.07	0.588
original_glrlm_LongRunHighGrayLevelEmphasis	6413.82 ± 5129	7259.89 ± 4796.66	0.994
original_glrlm_LongRunLowGrayLevelEmphasis	0.03 ± 0.07	0.03 ± 0.11	0.719
original_glrlm_LowGrayLevelRunEmphasis	0.01 ± 0.01	0.01 ± 0.01	0.321
original_glrlm_RunEntropy	6.89 ± 0.64	7.01 ± 0.61	0.357
original_glrlm_RunLengthNonUniformity	36,993.99 ± 9660.69	38,071.55 ± 8999.26	0.069
original_glrlm_RunLengthNonUniformityNormalized	0.82 ± 0.14	0.83 ± 0.13	0.087
original_glrlm_RunPercentage	0.88 ± 0.11	0.90 ± 0.10	0.095
original_glrlm_RunVariance	0.28 ± 0.43	0.26 ± 0.59	0.762
original_glrlm_ShortRunEmphasis	0.92 ± 0.07	0.92 ± 0.07	0.113
original_glrlm_ShortRunHighGrayLevelEmphasis	5532.21 ± 4666.78	6239.28 ± 4356.67	0.976
original_glrlm_ShortRunLowGrayLevelEmphasis	0.00 ± 0.01	0.00 ± 0.01	0.162
original_glszm_GrayLevelNonUniformity	468.18 ± 123.28	447.00 ± 123.74	0.781
original_glszm_GrayLevelNonUniformityNormalized	0.02 ± 0.01	0.02 ± 0.01	0.273
original_glszm_GrayLevelVariance	1388.83 ± 1007.04	1575.18 ± 1050.35	0.687
original_glszm_HighGrayLevelZoneEmphasis	5945.45 ± 4826.46	6695.76 ± 4512.44	0.949
original_glszm_LargeAreaEmphasis	69.67 ± 225.12	227.45 ± 1642.76	0.013
original_glszm_LargeAreaHighGrayLevelEmphasis	12,476.00 ± 7380.85	14,578.55 ± 17,349.9	0.219
original_glszm_LargeAreaLowGrayLevelEmphasis	4.33 ± 18.45	23.47 ± 179.94	0.006
original_glszm_LowGrayLevelZoneEmphasis	0.00 ± 0.01	0.00 ± 0.01	0.244
original_glszm_SizeZoneNonUniformity	22,268.19 ± 11,103.95	23,395 ± 10,680.69	0.170
original_glszm_SizeZoneNonUniformityNormalized	0.61 ± 0.17	0.62 ± 0.16	0.186
original_glszm_SmallAreaEmphasis	0.79 ± 0.12	0.81 ± 0.11	0.149
original_glszm_SmallAreaHighGrayLevelEmphasis	5360.88 ± 4462.06	6022.88 ± 4164.8	0.947
original_glszm_SmallAreaLowGrayLevelEmphasis	0.00 ± 0.00	0.00 ± 0.00	0.170
original_glszm_ZoneEntropy	7.71 ± 0.31	7.79 ± 0.30	0.622
original_glszm_ZonePercentage	0.67 ± 0.23	0.69 ± 0.21	0.068
original_glszm_ZoneVariance	65.35 ± 220.24	223.42 ± 1635.93	0.012
original_gldm_DependenceEntropy	7.84 ± 0.48	7.94 ± 0.48	0.476
original_gldm_DependenceNonUniformity	21,414.15 ± 8701.89	22,337.35 ± 8387.78	0.208
original_gldm_DependenceNonUniformityNormalized	0.43 ± 0.17	0.45 ± 0.17	0.208
original_gldm_DependenceVariance	1.21 ± 1.26	1.06 ± 1.14	0.042
original_gldm_GrayLevelNonUniformity	1220.12 ± 1123.5	1084.80 ± 1108.27	0.248
original_gldm_GrayLevelVariance	1343.93 ± 1003.18	1532.64 ± 1048.85	0.633
original_gldm_HighGrayLevelEmphasis	5605.65 ± 4740.08	6331.07 ± 4425.20	0.994
original_gldm_LargeDependenceEmphasis	5.65 ± 5.46	5.07 ± 5.33	0.133
original_gldm_LargeDependenceHighGrayLevelEmphasis	11,425.26 ± 7930.90	13,104.71 ± 7636.59	0.951
original_gldm_LargeDependenceLowGrayLevelEmphasis	0.14 ± 0.36	0.13 ± 0.42	0.741
original_gldm_LowGrayLevelEmphasis	0.01 ± 0.01	0.01 ± 0.01	0.399
original_gldm_SmallDependenceEmphasis	0.61 ± 0.22	0.63 ± 0.21	0.090
original_gldm_SmallDependenceHighGrayLevelEmphasis	4552.78 ± 4029.25	5112.5 ± 3752.87	0.964
original_gldm_SmallDependenceLowGrayLevelEmphasis	0.00 ± 0.00	0.00 ± 0.00	0.075

Values are expressed as the mean ± SD.

**Table 4 diagnostics-13-01863-t004:** Performance summary of algorisms used in the present study.

Model	Accuracy	Sensitivity	Specificity	F1 Score	AUC
SVM	0.52 (0.44–0.56)	0.52 (0.45–0.62)	0.52 (0.37–0.60)	0.49 (0.46–0.54)	0.53 (0.44–0.56)
LGBM	0.59 (0.52–0.69)	0.57 (0.41–0.79)	0.60 (0.29–0.89)	0.56 (0.52–0.60)	0.59 (0.52–0.67)
XGB	0.59 (0.48–0.69)	0.55 (0.42–0.69)	0.61 (0.40–0.86)	0.55 (0.44–0.64)	0.56 (0.49–0.68)
XGB based on RF	0.61 (0.50–0.67)	0.59 (0.46–0.66)	0.62 (0.38–0.76)	0.58 (0.51–0.62)	0.61 (0.51–0.66)
CatBoost	0.63 (0.56–0.69)	0.70 (0.52–0.86)	0.58 (0.44–0.80)	0.63 (0.56–0.74)	0.60 (0.57–0.70)
iRF	0.76 (0.70–0.80)	0.69 (0.62–0.77)	0.83 (0.65–0.94)	0.73 (0.71–0.76)	0.77 (0.73–0.83)

SVM, support vector machine; LGBM, light gradient boosting machine; XGB, extreme gradient boosting; RF, random forest; CatBoost, categorical boosting; iRF, improved random forest; AUC, area under the receiver operating characteristic curve.

**Table 5 diagnostics-13-01863-t005:** Confusion matrix for the prediction of all methods.

Model	TP	TN	FP	FN
SVM	74	89	82	69
LGBM	82	102	69	61
XGB	79	105	66	64
XGB based on RF	84	106	65	59
CatBoost	100	99	72	43
iRF	98	142	29	45

TP, True Positive; TN, True Negative; FP, False Positive; FN, False Negative; TP, True Positive; TN, True Negative; FP, False Positive; FN, False Negative; SVM, support vector machine; LGBM, light gradient boosting machine; XGB, extreme gradient boosting; RF, random forest; CatBoost, categorical Boosting; iRF, improved random forest.

## Data Availability

The datasets used and/or analyzed during the current study are available from the corresponding author upon reasonable request.

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
