# Peer review of "Machine Learning Assisting the Prediction of Clinical Outcomes following Nucleoplasty for Lumbar Degenerative Disc Disease"

_diagnostics, 2023, doi:10.3390/diagnostics13111863_

Round 1
Reviewer 1 Report (Previous Reviewer 2)
Thank you for answering my questions
Author Response
Dear Reviewer,
We would like to express our sincere gratitude for your time and valuable feedback on our manuscript titled "Machine Learning Assisting the Prediction of Clinical Outcomes Following Interventional Intradiscal Therapy for Lumbar Degenerative Disc Disease." Your thoughtful and constructive comments have significantly contributed to the improvement of our work's quality.
We truly appreciate the effort and expertise you have dedicated to reviewing our manuscript. Your insights have helped us refine our research and address important aspects of our study. We are grateful for your suggestions, which have undoubtedly enhanced the clarity and impact of our findings.
Once again, we would like to thank you for your valuable contribution. We are honored to have received your feedback, and we are committed to incorporating your suggestions to further strengthen our manuscript. We look forward to submitting the revised version and continuing this collaborative process with you.
Thank you again for your time and support.
Reviewer 2 Report (Previous Reviewer 1)
This manuscript has been revised based on review's comment. I think that this paper could be accepted, if the author improves their language quality of the paper.
The author improves their language quality of the paper.
Author Response
Dear reviewer,
Thank you for your valuable time and insightful comments on our manuscript titled "Machine Learning Assisting the Prediction of Clinical Outcomes Following Interventional Intradiscal Therapy for Lumbar Degenerative Disc Disease." We greatly appreciate your feedback, as it has played a vital role in enhancing the quality of our work.
We are pleased to inform you that we have arranged for the manuscript to undergo professional English editing, and we will provide the edited version as an attachment along with this response. We will also privide editing certificate as well. We believe that this editing process will further enhance the clarity and readability of our manuscript.
Once again, we express our gratitude for your thoughtful and constructive comments. Your input has been immensely beneficial in refining our research. We eagerly await your continued guidance and support as we work towards the publication of our study.
This manuscript is a resubmission of an earlier submission. The following is a list of the peer review reports and author responses from that submission.
Round 1
Reviewer 1 Report
This paper develops a ML–based model for predicting pain improvement after LNP for patients with LDDD, and the iRF ML model achieves the highest accuracy for predicting pain improvements following LNP in patients with LDDD. The experimental results look fine. However, it is a kind of being hard to understand what the proposed idea is useful, the innovation is not clear and thesis writing is not standardized. Therefore, there are quite a few points to be figured out.
1. Introduction, the authors need to emphasize the novelty and advantages of the proposed idea, and provide some more recent literature review in the introduction section.
2. It can be seen from previous researches that random forest algorithm has certain advantages compared with other algorithms. Is the author's improvement of random forest algorithm just the parameters optimization? Has the optimization algorithm been added?
3. It is recommended to include the CatBoost model in the comparison model. CatBoost is a better algorithm than XGBoost and LightGBM in terms of algorithm accuracy.
4. The description of the time required for network training should be added in Section 2.6.
5. In Section 4, as a comparison experiment, the confusion matrix for other methods should also be added instead of that for only iRF method.
6. The conclusion is too simple. The limitations of the proposed method and the prospect of future work should be given in conclusions.
Reviewer 2 Report
1. The serial number of the formula should be clearly indicated in lines 233-238.
2. Figure 4 is very blurry and cannot be seen clearly.
3. More details about the Mimics processing should be provided in the method part.
4. The conclusion is too simple without specific data support from line 365.
5. What is the innovation of this article? It should be noted in the introduction part.
6. How many patients were conducted from the method part?
7. What are the advantages of identifying this disease over traditional identification methods? This should be discussed in the discussion. What is the limitation of this study?
8. What about the pixels of the image from the method part?
9. How to deal with those blurrier areas in the method?
10. How do soft bone and porosity affect the results?
11. Why does the author use the LDDD method? what is the advantage compared with other methods?
132What is the reason to use the radiomic feature analysis?
Reviewer 3 Report
I have carefully read this manuscript and have some comments.
1. Although the article is well-designed, the small volume and low heterogeneity of the sample on various aspects is questionable.
2. Besides, I did not see the specific limitations of the research presented in the framework concerning specifically machine learning and so-called artificial intelligence.
3. The manuscript does not specify MRI machines and their field power (0.5-1-3, etc.). This also introduces inaccuracies in the training of artificial intelligence, as well as using only T2WI, since more than one mode is used for the standard spinal protocol.
В этом В связи с этим я не рекомендую эту статью для публикации в журнале, потому что данных, выборки и показателей недостаточно, чтобы судить о ML.